# Defect Detection and Characterization in Concrete Based on FEM and Ultrasonic Techniques

**DOI:** 10.3390/ma15228160

**Published:** 2022-11-17

**Authors:** Jeongnam Kim, Younho Cho, Jungwon Lee, Younghwan Kim

**Affiliations:** 1Graduate School of Mechanical, Pusan National University, Pusan 43241, Republic of Korea; 2School of Mechanical Engineering, Pusan National University, Pusan 43241, Republic of Korea; 3Plant Maintenance & Engineering Department, Korea Hydro & Nuclear Power Co., Ltd., Gyeongju-si 38120, Republic of Korea; 4Institute of Nuclear Safety and Management, Pusan National University, Pusan 43241, Republic of Korea

**Keywords:** concrete, FEM simulation, ultrasonic non-destructive techniques, crack depth evaluation

## Abstract

In order to estimate the crack depth in concrete using time-of-flight, finite element analysis and experiments were performed on non-cracked concrete blocks and 45 mm and 70 mm vertical cracks. As a result of measuring the time-of-flight change by changing the positions of the transmitter and receiver, it was confirmed that the finite element analysis results agreed with the experimental results, and high accuracy was confirmed by various formulas for calculating the depth of defects using the obtained experimental measurements for comparison. In addition to the verification of the simulation and experimental theory, research was conducted through actual field cases, and methodologies for crack detection and depth evaluation for concrete structures were presented, and furthermore, the expected effects of improving the soundness and safety of structures were shown.

## 1. Introduction

Compared to other structural materials, concrete structures have superior strength, durability, water tightness, steel protection performance, etc. They also have the advantage of being pourable without being greatly constrained by shape. However, there are also vulnerabilities that are prone to cracks, which may occur due to insufficient safety inspection and maintenance as the period of use elapses, due to design errors or defects in construction. If left unattended, cracks in concrete can affect the safety of the structure and result in crack growth due to several factors, including salt penetration. It is very important to manage cracks in order to secure the integrity of a concrete structure. If not managed through periodic inspection and proper maintenance, cracks can grow and lead to leakage or collapse.

One method for evaluating cracks in concrete structures with ultrasound is directly examining them with a pulser and receiver at opposite sides of the structure [1]. However, this method has a disadvantage in that it is difficult to inspect both sides of thick concrete structures due to the technical problems of ultrasonic scattering and ultrasonic generation. It may not be applied to features with restricted access or installed liners that restrict double-sided use. High-risk-rated structures are very thick, and some have metal liners installed, which often make double-sided ultrasound examination impossible [2,3,4].

The most important factors in the concrete crack inspection are the crack width, length, and depth [5,6,7,8]. Cracks visible on a concrete surface can be visually measured in length, but not in depth. Furthermore, in the case of microcracks that cannot be visually confirmed, other appropriate inspection methods are required. Much research has been done on the method of positioning a transducer on both sides of a specimen and directly determining the Ultrasonic Pulse Velocity (UPV) and the factors influencing it. A standard for measuring the velocity using direct transmission has been developed, but there is less information about indirect transmission. In general, indirect transmission is used when only one side of a concrete structure can be accessed [9,10,11].

According to the British Standard for UPV Measurement of Concrete (BS 1881), the indirectly transmitted UPV is 5–20% lower than the direct transmission velocity, which mostly depends on the concrete quality. Jones [12] describes the difference between direct and indirect velocity. Because a transducer is placed on the side of the concrete, the signal arriving at the front is too small to be missed and seems to be delayed. The BS 1881 standard recommends an indirect velocity measurement procedure, using the relationship between the transducer spacing and the propagation time by repeating the propagation-time measurement while increasing the distance between the transducers.

The American Standard for UPV Measurement of Concrete (ASTM C 597) does not recommend measuring the UPV by indirect transmission unless accessible from one surface of the material. It indicates that indirect transmission is a characteristic of the layer close to the surface. This is because only the value obtained from the direct transmission can be expressed. Currently, there is no ASTM standard for indirect UPV measurement.

M. Sansalone et al. proposed a technique for determining the thickness of concrete elements and finding defects in concrete through P-wave velocity measurement technology, by applying indirect transmission in concrete. Two transducers were placed on a concrete surface, and an impact was generated at a point along a line drawn between the axes of the two transducers. The UPV was calculated as the ratio of the distance between the transducers and the ultrasonic propagation time between the transducers, and recommendations were made for the transducer spacing when determining the UPV [13]. Indirect UPV measurements were also compared with the direct measurements performed by Popovics et al. using a similar procedure used by Sansalone, Lin, and Streett, and the equivalence of indirect UPV and direct UPV was verified for homogeneous materials [14].

N. M. Sutan et al. conducted a comparative experiment with direct and indirect methods of UPV to detect concrete defects. Through this, a relational expression that can find voids in concrete and estimate the depth was verified. After 3, 7, 14, and 28 days elapsed, five Grade-40 concrete specimens were fabricated, and the accuracy of the two methods was confirmed. As a result, the direct transmission showed more convenient and satisfactory results in the sensitivity of judging the location of a defect than the indirect transmission, but the ability to determine the depth of the void was lacking. Therefore, only indirect transmission was possible when both the depth and location of the void had to be determined, and the accuracy was 60–99% depending on the curing period [15]. However, these methods have the disadvantage of having a large error, even when tested in theory or used in an industrial field.

The aims of this study were the following. (1) Simulation was done using the UPV method for concrete blocks as a new approach that could not be confirmed in previous studies. (2) We compared and verified the accuracy of measurements obtained through theoretical and practical experiments. (3) We conducted research on actual field cases as well as verification of the simulation and experimental theory. (4) We present methodologies for crack detection and depth evaluation for concrete structures. We confirmed their applicability in actual sites.

## 2. Materials and Methods

This section covers some techniques used in crack detection, depth measurement studies, and results that were confirmed through experiments. It summarizes the composition of ultrasonic inspection equipment, transducer and frequency determination, concrete specimen experiment, simulation for concrete ultrasonic examination modeling, and examination results for cracks in actual structures.

A compression wave is generated by an electro-acoustic transducer in contact with a concrete surface. This pulse passes through the concrete and the second transducer, which is separated from the transmitting transducer by a distance (L). It receives the pulse and converts it into electrical energy. The propagation time (T) is measured, and the pulse velocity V is calculated with Equation (1) by dividing L by T:(1)$$V=\sqrt{\frac{E\left(1-\mu \right)}{\rho \left(1+\mu \right)\left(1-2\mu \right)}}$$

E is the dynamic modulus of elasticity, μ is the dynamic Poisson’s ratio, and ρ is the density [9]. The pulse velocity (V) is calculated according to the following equation.
(2)$$V=\frac{L}{T}$$

V is the pulse velocity (m/s or mm/μs), L is the center-to-center distance of the sender/receiver surfaces (m or mm), and T is the propagation time (s or μs). Because concrete saturation affects the pulse rate, this factor must be considered when measuring the velocity. The pulse rate of saturated concrete can be as large as 5% and is less sensitive to changes in the relative quality of concrete.

The accuracy of the measurement may vary depending on the tester’s ability to measure the propagation distance and the performance of the equipment. The strength of the received signal and the measured propagation time are affected by how well the transmitter and the receiver adhere to the concrete surface. The strength of the received signal is affected by the movement distance of the pulse between the transmitter and the receiver and the degree of cracking or deterioration of the concrete.

The pulse velocity inside iron reaches twice the velocity in concrete, so when the pulse velocity is measured near reinforcement, the ultrasonic propagation velocity may be greater than that in unsupported concrete of the same composition. Therefore, measurements near reinforcing bars parallel to the direction of propagation of the pulses should be avoided where possible [16]. Lastly, the ultrasonic propagation speed may differ depending on the type of aggregate. It was found that the ultrasonic propagation speed was slower in concrete containing round-cornered aggregate than in concrete containing pulverized aggregate [17].

Sutan and Meganathan obtained UPV values by indirect penetration through one side of the concrete. Using this, we evaluated the accuracy of defect detection and depth measurement with a relational expression for calculating the depth of a void. The presence of the defect could be confirmed 100% of the time, but it showed accuracy in the range of 60–99% in evaluating the depth of the defect. Equation (3) below is a relational expression applied to check the depth of a void H, and Figure 1 shows a diagram of a crack detection experiment using an indirect transmission.
(3)$$H=\frac{D}{2}\sqrt{\frac{V_{S}-V_{C}}{V_{S}+V_{C}}}$$

V_S_ is the pulse velocity in sound concrete, V_C_ is the pulse velocity in defect concrete, and D is the distance at which the change of slope occurs. Although this method can detect the presence of cracks, the accuracy of the measured depth is poor.

The relational expression for evaluating the depth of the crack formed in the vertical direction from the surface studied by Leslie et al. can be derived as follows. In order to determine the depth of a crack, a transmitter and a receiver are placed on both sides at the same distance from the crack. When an ultrasonic wave arrives at the tip of a crack, the wave is diffracted and propagated as a spherical wave, arriving at the receiver placed on the opposite side. [18] Since the ultrasonic wave arrives by diffraction at the tip of the crack, the arrival time of the ultrasonic wave is determined by the depth H of the crack and the distance D between the transducers.

The arrival time (T_C_) of the ultrasonic wave is given by the following equation:(4)$$T_{C}=\frac{\sqrt{D^{2}+4H^{2}}}{V}$$
where V is ultrasonic wave velocity in the concrete. The arrival time T_S_ of the ultrasonic wave in the area where cracks do not exist is given by Equation (5) below. The depth of crack (H) can be obtained with Equation (6). This method of evaluating the depth of cracks is known as the T_C_-T_S_ method [19].
(5)$$T_{S}=\frac{D}{V}$$
(6)$$H=\sqrt{{\left(\frac{T_{C}}{T_{S}}\right)}^{2}-1}$$

Equation (6) assumes that the speeds of ultrasonic waves in a concrete surface and inside are the same, but in reality, the speed of a surface wave and that of an internal wave in concrete are different, so the T_C_-T_S_ method is not accurate [20]. Since it is obtained from simple geometric calculations, the ultrasonic delay inside the transducer is not considered. In particular, when the delay time inside the transducer is long, the depth of the crack determined by Equation (6) shows a large difference from the actual value.

## 3. Concrete Crack Modeling and Simulation

When cracks growing in the thickness direction exist on a concrete surface, a simulation can be performed to determine whether the crack detection and depth could be quantified using the UPV of one surface. For this, a concrete block with different crack depths was modeled by Onscale (OnScale 1.30.11.0, ANSYS, Redwood, CA, USA) which is a cloud engineering simulation platform, and the easiness of UPV inspection using two different frequencies was compared. Theoretically, the ultrasonic time-of-flight (TOF) increases proportionally as the distance between the transducers (Olympus, Tokyo, Japan) increases when there are no cracks and increases as the ultrasonic wave travels through the cracks.

After modeling a 2D-shaped concrete block, the general physical properties of the concrete block (see Table 1) were input, and the TOF was measured while increasing the distance between the transmitter and the examiner at regular intervals. Then, the accuracy of a previously known crack-evaluation relationship was confirmed by using the distance between the transducers with the cracks between them and the increased TOF or UPV change.

The physical shape of the concrete block model was simulated in 2D as 600 mm (length) × 150 mm (height). The models were composed of three types. The first and second ones had vertical cracks of 70 mm and 45 mm in the center of the block, respectively, and the other had no cracks. The three shapes of concrete blocks are shown in Figure 2 below. In addition, a total of 10 models were constructed by subdividing them according to the frequency and the positions of the transmitter and receiver as shown in Table 2.

ASTM C597 requires that a transmitter with a resonant frequency in the range of 20 to 100 kHz be used for an ultrasonic propagation speed test in concrete. Therefore, the ultrasonic frequency for the main simulation was selected as an approximately intermediate value of 65 kHz. In order to confirm the effect of the difference in the distance between the crack and the transducer on evaluating the depth of the crack, the ultrasonic pulse speed was measured by varying the distance between the transmitter and the crack for each depth of the crack.

As shown in Figure 3, the TOF is the same up to Transmitter 1, but at Receiver 2 behind the crack, the ultrasonic wave is diffracted at the tip of the crack and propagates. It can be seen that the TOF increases as the movement distance of the pulse increases. The collected signals corresponding to the positions of receiver 1 and receiver 2 are shown in Figure 4.

The propagation time (TOF) of the ultrasonic pulse was measured based on the first peak point of the pulse that appeared when the pulse reached the receiver, as shown in Figure 5. When the ultrasonic pulse oscillating from the transmitter travels and encounters a crack, it does not pass through the crack but is diffracted at the tip of the crack and propagates to the receiver. This extends the movement distance of the ultrasonic wave. The TOF increases when passing through the crack due to the increase in movement distance.

The TOF change with the difference in the distance between the transducer and the crack was compared under the condition of cracks of the same size. Cracks No. 1 and No. 3 are located at a distance of 270 mm from the transmitter, and No. 2 and No. 4 are located at a distance of 150 mm. In the section without cracks, TOF increased at a certain rate, but when passing through sections with cracks of 70 mm and 45 mm, TOF showed a larger value (see Figure 6 and Figure 7). However, after passing through each crack, it was confirmed that the cumulative TOF at a distance of 420 mm between the transmitter and the examiner was at a similar level. Through this, it was confirmed that the separation distance between the transducer and the crack did not affect the cumulative TOF.

With the crack in the center, the crack was evaluated using the TOF of each section by the examinee’s position under the condition where the distance before and after the crack was the same. For the speed of the uncracked part, the TOF of No. 5 was applied. For the cracked part, we used the TOF confirmed by the simulation of the corresponding model.

As shown in Table 3, Table 4, Table 5 and Table 6, the depth of the crack was calculated with an accuracy of 29.7 to 153.9% depending on the depth of the crack and the distance between the probes. It was possible to obtain the result that the evaluated depth of the crack increased as the distance between the probes increased. However, a crack value larger than the crack size (depth) was calculated when the interval was out of a certain range, so it was found that cracks could be evaluated relatively accurately at a probe spacing that intersects the actual value.

## 4. Experiment with Concrete Specimen and Results

### 4.1. Experiment Setup

The ultrasonic pulse rate is affected by factors such as the cement type [21], aggregate type and size, water-to-cement ratio, distance between transducers, admixture, transducer location, and concrete lifetime [22]. Therefore, for the evaluation of cracks in the concrete structures, concrete specimens were prepared with a mixing ratio similar to that of the actual high-risk facility structures, and the experimental equipment was also configured in the same way as that applicable to the actual inspection. Table 7 shows the concrete mixing ratio.

For the experiment, as shown in Figure 8, three concrete blocks with a size of 600 mm (L) × 200 mm (W) × 150 mm (H) and the same mixing ratio were manufactured. One of these is a block with no cracks on the surface and was used to check the UPV in a defect-free area. The other two have vertical cracks machined in the center with a width of 0.5 mm and depths of 45 mm and 70 mm, respectively. The accuracy of the relational expression for measuring depth was confirmed through the difference in propagation time of ultrasonic pulses when passing through cracks. The optimal test conditions that can be applied to the actual test conditions were confirmed by reflecting the ultrasonic delay inside the transducer.

The transmitter and the receiver were separated by a distance. The surface of the concrete block was marked at intervals of 60 mm so that a constant value could always be obtained when measuring the ultrasonic speed. A diagram of the experimental apparatus is shown in Figure 9.

A measurement of the ultrasonic pulse propagation time on the concrete surface was performed with the same conditions as the simulation model. In a concrete block with 45 mm and 70 mm cracks, the transmitter was positioned at 150 mm and 270 mm away from the crack, and the propagation time (TOF) of the pulse was measured. The propagation time and speed of ultrasonic pulses were checked on the block surface up to a distance of 540 mm, the maximum range that can be measured at intervals of 60 mm. In order to improve the reproducibility and accuracy of the experimental results, the ultrasonic pulse propagation time was measured a total of five times for each condition, and the average value was used to apply the known relational expression to evaluate the depth of the crack.

### 4.2. Experiment Setup

In experiments and simulations, the time of propagation (TOF) of ultrasonic pulses was determined by checking the first peak of the first cycle as in the previous experiments. [23]. In order to clarify the TOF, it is necessary to identify the first peak point accurately Ultrasound has the advantage of better resolution as the frequency increases, but it has the disadvantages of a short wavelength, large attenuation, and low propagation power at higher frequencies.

Since various materials such as aggregate and cement are mixed inside the concrete, the damping (scattering and absorption) is large. ASTM C597 requires using a resonant frequency in the range of 20 to 100 kHz, so it is easy to identify the first peak by selecting 65 kHz, which is within the frequency range required by ASTM C597. With 110 kHz, the resolution can be expected to be a little better in the simulation and experiment, so it was compared with 65 kHz. Signals using 65 kHz and 110 kHz are shown in Figure 10. When 110 kHz is used, it is difficult to distinguish the signals. This is in line with ASTM C597. Higher frequencies will only be applicable to concrete with low attenuation.

As shown in Table 8 and Figure 11, except for the crack section, the TOF increased at a constant slope as the transducer spacing increased in the remaining sections. Compared with the simulation results in Figure 6, the TOF value of No. 1 was higher than that of No. 3 and No. 5. However, unlike the simulation results that showed differences in Nos. 3 and 5, measurements were similar in the actual experiment. In addition, in the case of the TOF measurement value after cracking, it was possible to measure an approximate value when there was a crack in the simulation, but in the case of the actual experiment, even if there was a crack, the depth was measured.

The depth of a vertical crack was obtained when the TOF value measured experimentally was substituted into the relational expression from Sutan and from Leslie and Cheesman. The accuracy of the two relations was confirmed. In addition, only the test results of 65 kHz were used to evaluate the depth of cracks. In relation to Sutan and Meganathan, the crack evaluation was conducted with different transducer spacing as in the simulation, and the accuracy of the crack evaluation was confirmed.

### 4.3. Comparison of Sutan and Meganathan’s Equations with Simulation Results

In the case of Experiment 1, as shown in Figure 12, the crack depth was calculated with an accuracy of 24.9–76.3% depending on the depth of the crack and the spacing between the probes with the 70 mm crack condition. As with the simulation conditions, it was confirmed that the farther the transducer spacing is, the closer the crack-depth evaluation result is to the entered crack size (depth), and the higher the accuracy is. With the 45 mm crack condition, the depth of the crack could not be evaluated when there was a crack, there was no TOF difference between the probes, and the gap between the transducers was large. However, the accuracy was 7.4–7.5%, which is judged to be a level that can only confirm the presence or absence of cracks.

When the relationship between Sutan and Meganathan was applied, it was confirmed that the accuracy increased as the distance between the transducers increased in both the experiment and the simulation with the 70 mm crack condition. However, since all the results were evaluated with a value smaller than the actual crack depth, it is expected that the accuracy can be improved if the probe is tested with a longer distance between the probes.

### 4.4. Comparison of Leslie and Cheesman’s Equations with Simulation Results

In the case of experiment No. 1, as shown in Figure 13, with the 70 mm crack condition, the depth of the crack was calculated with an accuracy of 75.4 to 158.7% depending on the depth of the crack and the distance between the probes. When a crack exists at a point 270 mm away from the transmitter, the depth of the crack was evaluated as the distance between the transducers increased. The value was calculated. In the case of a crack at 150 mm from the transmitter, the accuracy of the crack-depth evaluation was 80.1 to 102.6%, indicating a relatively even result.

Therefore, when the concrete surface for crack evaluation is wide, it is judged that the accuracy of crack-depth evaluation can be improved by varying the gap between the transmitter and the examiner to evaluate the crack depth in various ways. However, since this experiment evaluated only the accuracy of the crack depth with a single condition of 70 mm, the accuracy, when the crack depth was different, was unknown, so the conversion of the crack evaluation relational expression was not considered.

In the 45 mm crack condition, there was no difference in TOF from the case without cracks, so the depth of the crack could be quantified only when using the TOF change values of the front and rear ends of the crack. However, the accuracy is at the level of 15.2 to 15.5%, which is judged to be a level at which only the presence or absence of cracks can be confirmed. When the relational expression for crack measurement was applied, using both simulation and experimental results, we could obtain closer evaluation results to the actual crack depth compared to the previous relational equations for crack evaluation.

With the 70 mm crack condition, both the experiment and the simulation showed the same tendency for the depth of the crack to increase as the distance between the transducers increased. Through this, it was found that in the crack evaluation using Leslie and Cheesman’s equations, it would be possible to evaluate the depth of cracks more precisely after determining the optimal transducer spacing. In the 45 mm crack condition, the depth of the crack could be evaluated in the simulation when the relationship between Sutan and Meganathan was applied. The accuracy was very low at 15.2 to 15.5%.

### 4.5. An Example of Field Measurement

The next inspection target was a surface where cracks can be visually confirmed on the floor near a site where a concrete storage facility is located. The concrete surface where cracks were confirmed was a floor area near a concrete building, which had nothing to do with the soundness of the structure, but there was no difference in detecting and evaluating cracks using the UPV method. Therefore, crack detection and crack-depth evaluation using UPV were performed.

The ultrasonic speed was measured a total of four times while the position of the examinee was spaced by 60 mm, and the transmitter was fixed in the same condition as the experimental conditions on the test piece (60 mm, 120 mm, 180 mm, 240 mm). As the accuracy was confirmed by the experiment, the ultrasonic propagation time was measured with the crack at 150 mm from the transmitter. A diagram of the ultrasonic inspection of the bottom part where cracks were confirmed is shown in Figure 14. In addition to the cracked part, the ultrasonic propagation time was additionally measured for the floor part without cracks in the vicinity. It was used to detect cracks and evaluate the depth.

As shown in Figure 15, the TOF of the first section from the transmitter was 63.2–68.0 μs. This result was increased compared to the result of about 50 μs in the experiment. This seems to be the result of the concrete composition ratio leading to a difference in the ultrasonic propagation speed inside.

In the section where cracks exist (120→180 mm), the difference between the same section and the same section without cracks was large. The crack showed a TOF increase of 72.9 μs, similar to the 63.5-μs increase in TOF in the same section of Experiment 2 with similar conditions. On the other hand, in the area without cracks, the TOF increase showed a large difference of 16.4 μs, so the existence of cracks could be easily confirmed even with the TOF change. When inspecting the concrete floor surface, the TOF change greatly increased past the crack and showed a similar shape to that of the test with the 70 mm crack condition. Considering that the TOF change of the 45 mm crack in the experiment was insignificant, it was estimated that the crack in the floor was deeper than at least 45 mm.

As shown in Table 9, when the relationship between Sutan and Meganathan was applied, the depth of the crack was estimated to be 23.9 to 61.0 mm. However, in Experiment 2 with similar conditions, the crack depth evaluated was found to be 19.3 to 25.7 mm. Therefore, it was possible to obtain a result that could be estimated to have a crack at the bottom of at least 70 mm.

As shown in Table 10, when Leslie and Cheesman’s equations were applied, the depth of the crack was estimated to be 129.9 to 225.6 mm. In the same conditions as Experiment 2, the depth of the crack was 56.1 to 66.2 mm, and the accuracy was 80.1 to 94.6%. As in the previous evaluation, the depth of the crack in the concrete floor could be estimated to be at least 70 mm. In particular, in the experiment, the accuracy of evaluating the depth of a crack by the change in TOF of 60 mm before and after the crack was 94.6%, which was relatively high. However, since it is judged that the accuracy of this quantitative evaluation requires experimental results with more diverse conditions, it seems reasonable to use it to estimate only the tendency of the depth of cracks.

## 5. Conclusions

Finite element analysis and experiments were performed on non-cracked concrete blocks and 45 mm and 70 mm vertical cracks to estimate the crack depth in concrete using ultrasonic propagation time. As a result of measuring the time-of-flight change by changing the positions of the transmitter and receiver converters, it was confirmed that the finite element analysis results agree well with the experimental results. The depth of the defect was calculated using various formulas using the measured values, and the accuracy was compared to show a high degree of similarity to the theoretical value. It also shows the results of applying the verified UPV method to cracks found in the actual field.

In this study, the theoretical crack length measurement method was verified through FEM(Finite Element Method) modeling and actual experiments. In addition, it was confirmed that the crack detection and depth evaluation method using ultrasonic pulse velocity in cracked concrete structures can be a very useful tool to maintain the soundness and safety of the structure.

## Figures and Tables

**Figure 1 materials-15-08160-f001:**
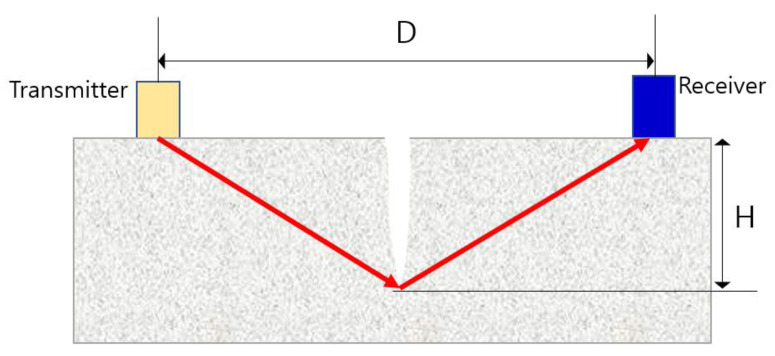
Crack perpendicular to the surface and arrangement of the transducers (T_C_-T_S_ method).

**Figure 2 materials-15-08160-f002:**
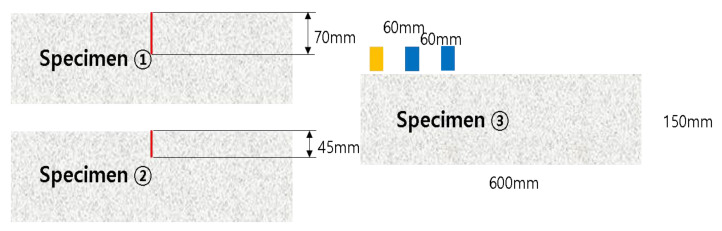
Concrete specimen model (2D).

**Figure 3 materials-15-08160-f003:**
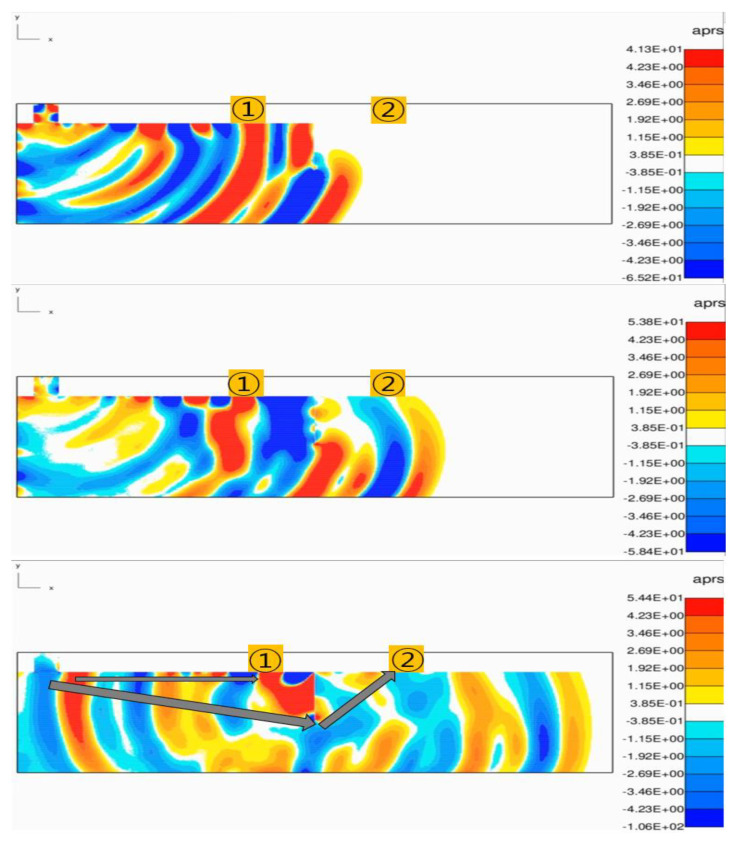
Onscale simulation result image No. 1 (70 mm crack).

**Figure 4 materials-15-08160-f004:**
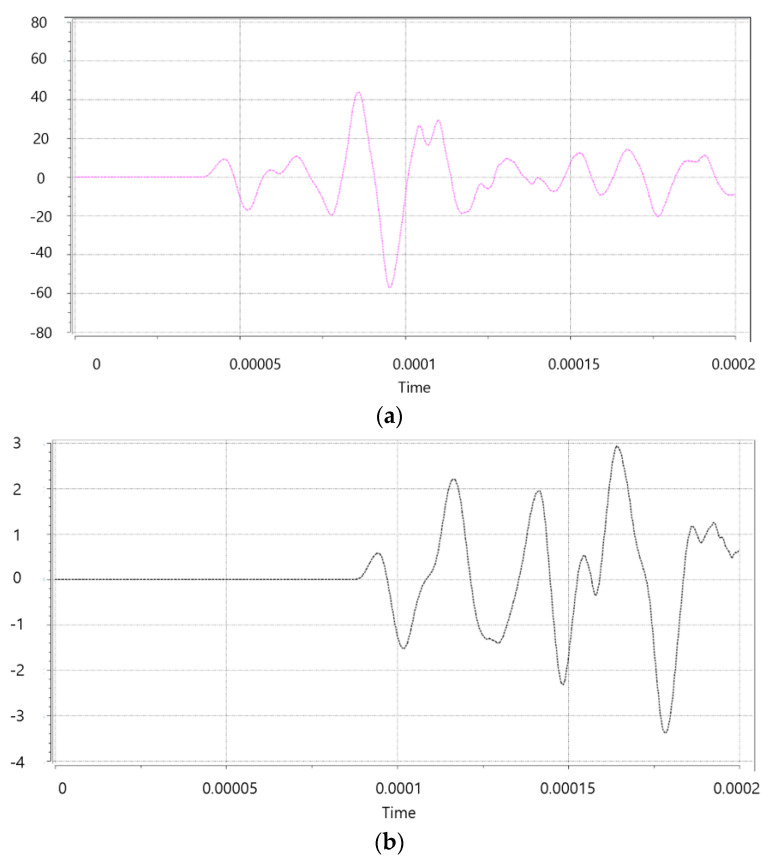
Signal waveforms (**a**) at Receiver 1 and (**b**) at Receiver 2.

**Figure 5 materials-15-08160-f005:**
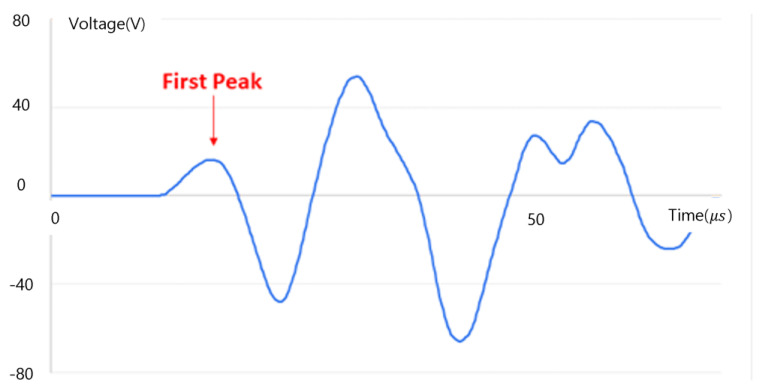
TOF measurement reference point (first peak).

**Figure 6 materials-15-08160-f006:**
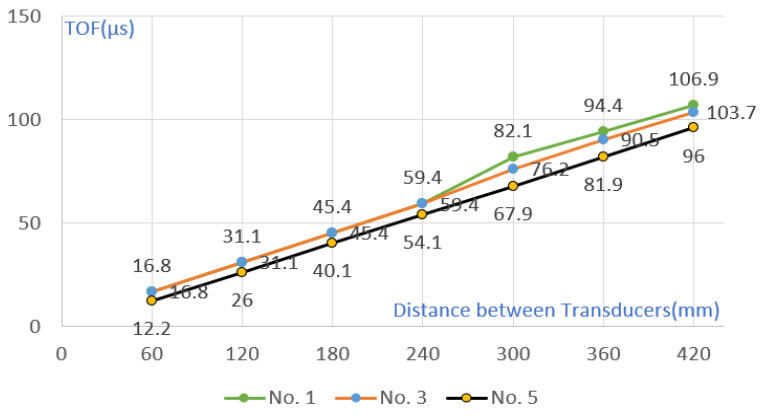
Comparison of TOF with 45 mm crack, 70 mm crack, and no-crack conditions.

**Figure 7 materials-15-08160-f007:**
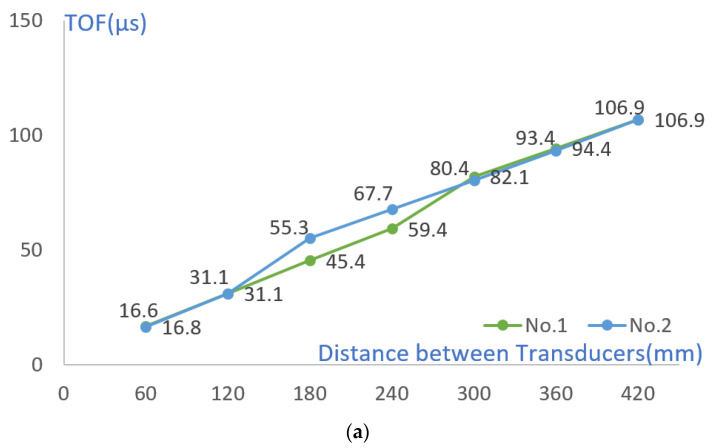
TOF according to the separation distance between the transmitter and the crack of (**a**) 70 mm (**b**) 45 mm.

**Figure 8 materials-15-08160-f008:**
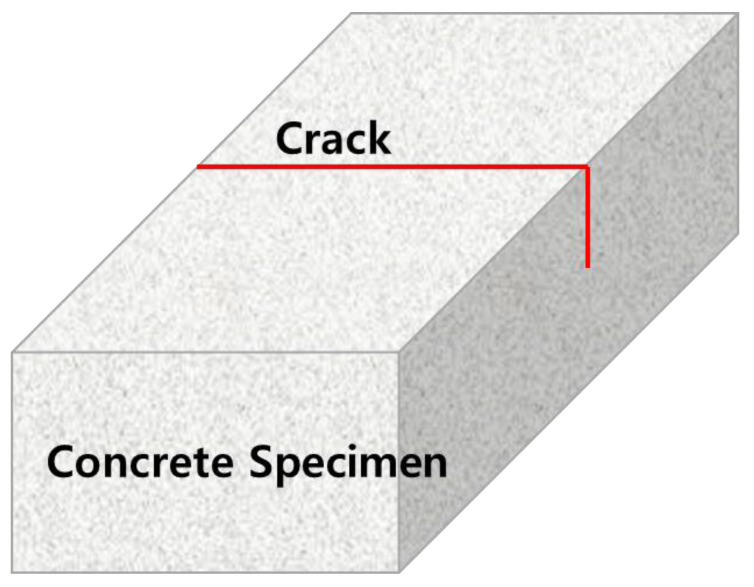
Concrete specimen with vertical cracks.

**Figure 9 materials-15-08160-f009:**
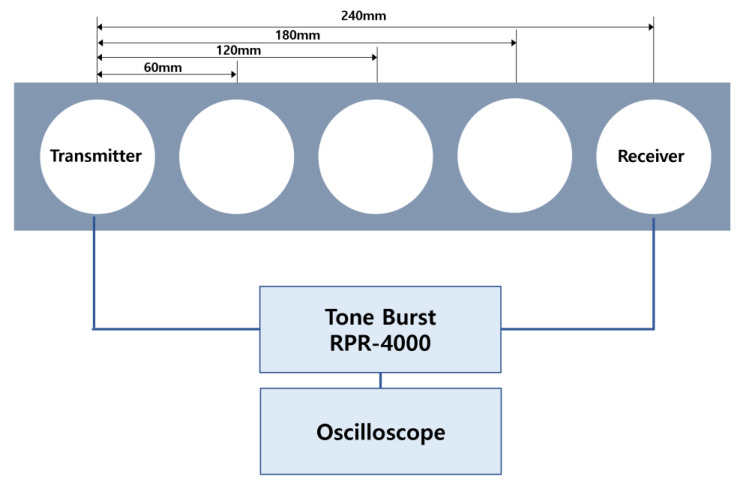
Diagram of ultrasonic wave velocity measurement experiment.

**Figure 10 materials-15-08160-f010:**
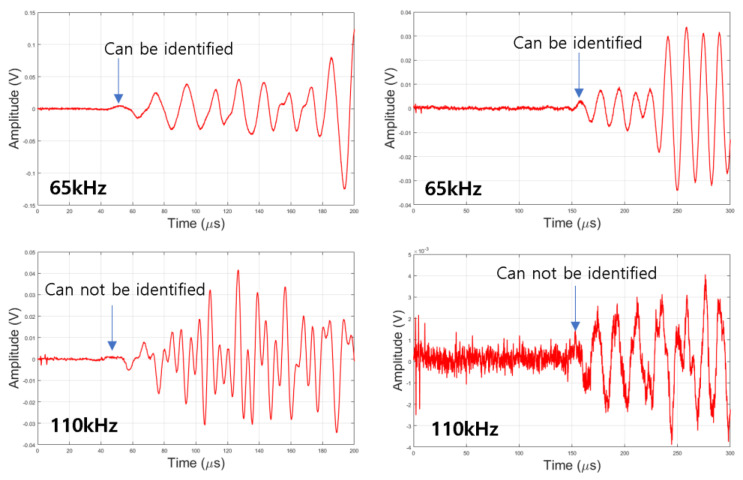
Received waveform.

**Figure 11 materials-15-08160-f011:**
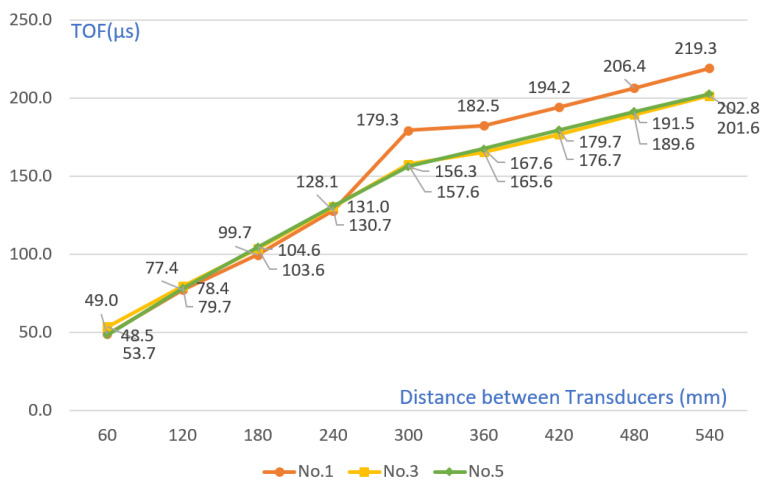
Comparison of TOF with 45 mm crack, 70 mm crack, and no-crack conditions in concrete specimen.

**Figure 12 materials-15-08160-f012:**
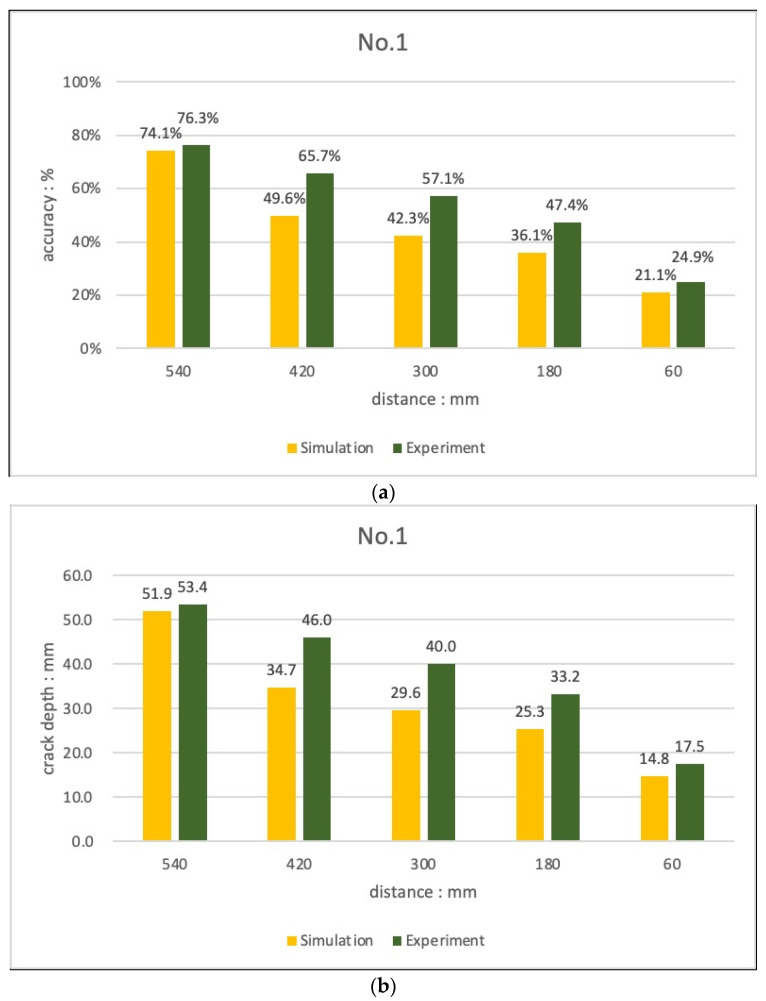
Comparison of crack evaluation results (Sutan and Meganathan’s equations, experiment No. 1) (**a**) 70 mm crack at 270 mm point (**b**) 70 mm crack at 150 mm point.

**Figure 13 materials-15-08160-f013:**
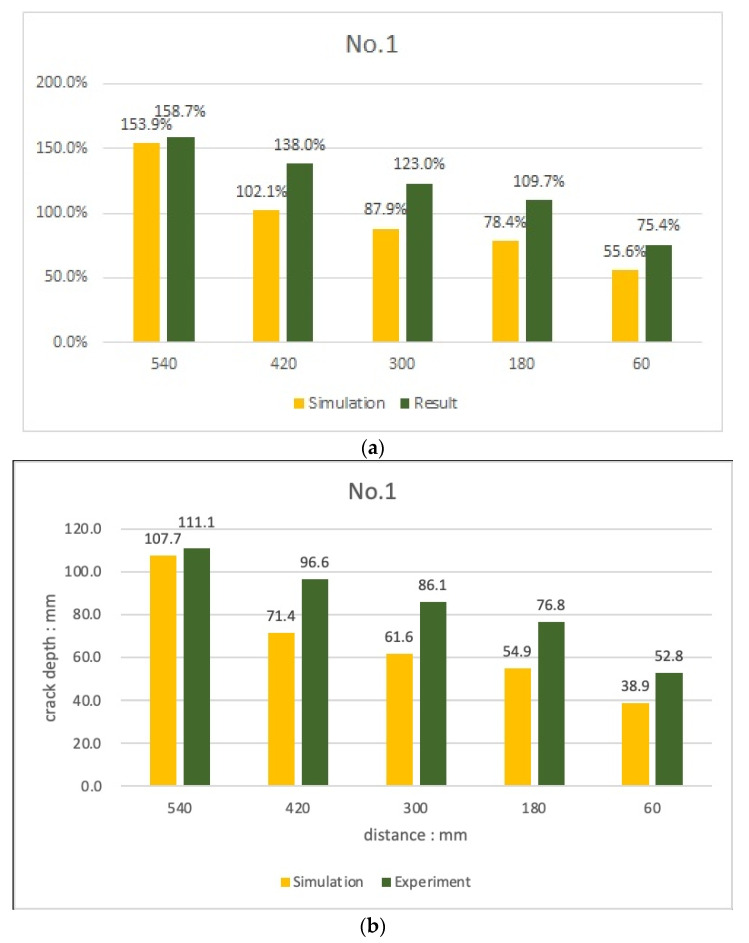
Comparison of crack evaluation results (Leslie and Cheesman’s equations, experiment No. 1) (**a**) 70 mm crack at 270 mm point (**b**) 70 mm crack at 150 mm point.

**Figure 14 materials-15-08160-f014:**
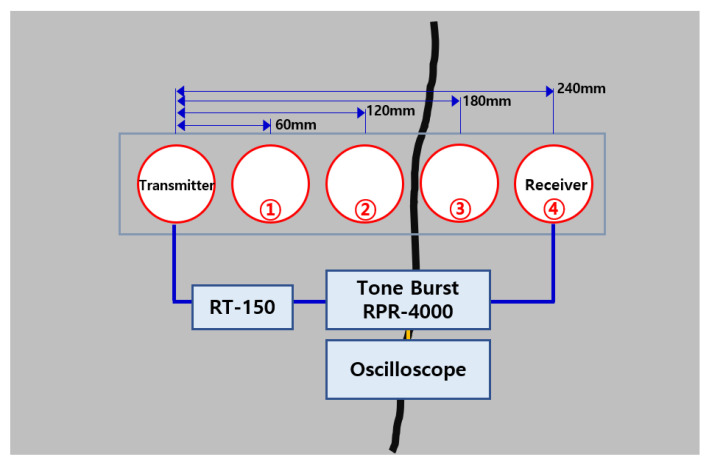
Concrete crack area for UPV measurement (bottom surface).

**Figure 15 materials-15-08160-f015:**
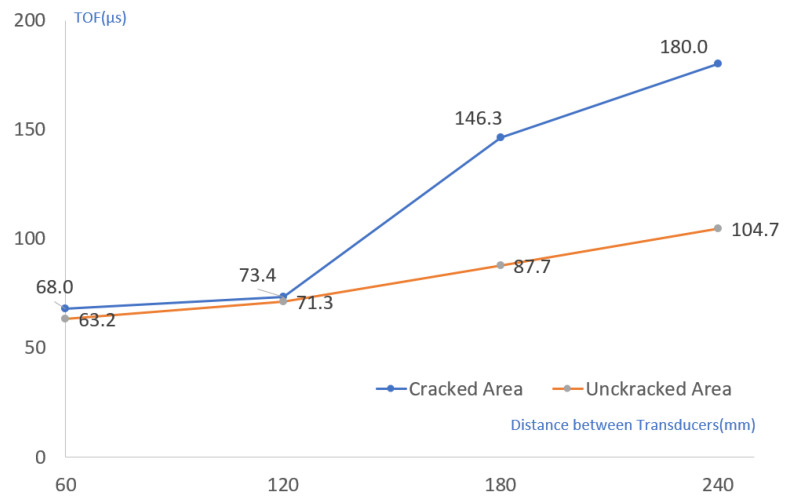
Comparison of ultrasonic pulse time-of-flight (TOF) for cracked and uncracked areas.

**Table 1 materials-15-08160-t001:** Material properties of the concrete.

Density	Velocity	
	Longitudinal	Shear
2369 kg/m^3^	4320 m/s	2450 m/s

**Table 2 materials-15-08160-t002:** List of models.

No.	Frequency	Depth of Crack	Distance between Transducers	No. of Measurement Time
1	65 kHz	70 mm	270 mm	9
2			150 mm	7
3		45 mm	270 mm	9
4			150 mm	7
5		No crack	-	9

**Table 3 materials-15-08160-t003:** Simulation TOF crack-depth evaluation result of case No. 1 (70 mm crack at 270 mm point).

Measurement Section (mm)	0→540	60→480	120→420	180→360	240→300
Transducer spacing (D, mm)	540	420	300	180	60
Uncracked part TOF ($T_{S}$, mm)	123.8	97.5	70.1	41.8	13.8
Cracks TOF ($T_{C}$, mm)	133.3	103.0	75.7	48.9	22.6
Crack-depth evaluation result (t, mm)	107.7	71.4	61.6	54.9	38.9
Accuracy (%)	153.9%	102.1%	87.9%	78.4%	55.6%

**Table 4 materials-15-08160-t004:** Crack-depth evaluation result with case No. 2 (70 mm crack at 150 mm point) simulation TOF.

Measurement Section (mm)	0→300	60→240	120→180
Transducer spacing (D, mm)	300	180	60
Uncracked part TOF ($T_{S}$, mm)	67.9	41.9	13.8
Cracks TOF ($T_{C}$, mm)	80.4	51.5	24.1
Crack-depth evaluation result (t, mm)	95.0	62.6	41.5
Accuracy (%)	135.7%	89.5%	59.3%

**Table 5 materials-15-08160-t005:** Crack-depth evaluation result of case No. 3 (45 mm crack at 270 mm point) simulation TOF.

Measurement Section (mm)	0→540	60→480	120→420	180→360	240→300
Transducer spacing (D, mm)	540	420	300	180	60
Uncracked part TOF ($T_{S}$, mm)	123.8	97.5	70.1	41.8	13.8
Cracks TOF ($T_{C}$, mm)	130.8	100.4	72.6	45.1	16.8
Crack-depth evaluation result (t, mm)	92.0	51.0	40.4	36.6	20.8
Accuracy (%)	131.4%	72.8%	57.7%	52.3%	29.7%

**Table 6 materials-15-08160-t006:** Crack-depth evaluation result of case No. 4 (45 mm crack at 150 mm point) simulation TOF.

Measurement Section (mm)	0→300	60→240	120→180
Transducer spacing (D, mm)	300	180	60
Uncracked part TOF ($T_{S}$, mm)	67.9	41.9	14.1
Cracks TOF ($T_{C}$, mm)	76.4	46.4	17.5
Crack-depth evaluation result (t, mm)	77.3	42.8	21.8
Accuracy (%)	110.4%	61.1%	31.1%

**Table 7 materials-15-08160-t007:** Concrete mixing ratio.

Water	Cement	Aggregate			Total
		Fine (Sand)	Coarse (3/4 Inch)	Coarse (1–1/2 Inch)	
6.66%	13.55%	29.11%	25.34%	25.34%	100%

**Table 8 materials-15-08160-t008:** TOF change value by crack size (depth).

Distance of Transducers (mm)	60	120	180	240	300	360	420	480	540
No. 1 (70 mm crack)	49.0	77.4	99.7	128.1	179.3	182.5	194.2	206.4	219.3
No. 3 (45 mm crack)	53.7	79.7	103.6	130.7	157.6	165.6	176.7	189.6	201.6
No. 5 (uncracked)	48.5	78.4	104.6	131.0	156.3	167.6	179.7	191.5	202.8

**Table 9 materials-15-08160-t009:** Crack depth evaluation using Sutan and Meganathan’s equation.

Measurement Distnace (mm)	60→240	120→180
Distance of transducers ($X_{0}$, mm)	180	60
Wave speed at uncracked ($V_{s}$, mm/μs)	4.3	3.7
Wave speed at cracked ($V_{d}$, mm/μs)	1.6	0.8
Crack depth evaluation result (t, mm)	61.0	23.9

**Table 10 materials-15-08160-t010:** Crack depth evaluation using Leslie and Cheesman’s equations.

Measurement Distance (mm)	60→240	120→180
Distance of transducers ($X_{0}$, mm)	180	60
Wave speed at uncracked ($V_{s}$, mm/μs)	41.5	16.4
Wave speed at cracked ($V_{d}$, mm/μs)	112.0	72.9
Crack depth evaluation result (t, mm)	225.6	129.9

## Data Availability

Not applicable.

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
