# Peer review of "Defect Detection and Characterization in Concrete Based on FEM and Ultrasonic Techniques"

_materials, 2022, doi:10.3390/ma15228160_

Round 1

Reviewer 1 Report

This paper proposed an evaluation method to detect cracks using ultrasonic pulse echo method where acoustic transducers were placed on one surface of a storage facility. Simulations and experiments were performed for verifications. In general, although the research in this paper is scientific, reviewers can hardly find the uniqueness of this paper, whether from the perspective of measurement methods, objects, or the purpose of using ultrasonic pulse velocity for crack detection and depth evaluation.

Some other questions:

L40, "The most important factors in concrete crack inspection are crack width, length and 40

depth." the width of the crack may be of low importance, while the direction of the crack defect is more important.

How to determine the crack length in this paper?

Author Response

Authors feel grateful for the invaluable comments from the reviewer.

“In general, although the research in this paper is scientific, reviewers can hardly find the uniqueness of this paper, whether from the perspective of measurement methods, objects, or the purpose of using ultrasonic pulse velocity for crack detection and depth evaluation.”

Finite element analysis and experiments were performed on uncracked concrete blocks and 45 mm and 70 mm vertical cracks to estimate the crack depth in concrete using ultrasonic propagation time. Simulation using the UPV method for concrete blocks is a new approach that could not be confirmed in previous studies. As a result of measuring time-of-flight changes by changing the positions of the transmitter and receiver, it was confirmed that the finite element analysis results agree well with the experimental results, and various formulas for calculating the depth of defects using the obtained experimental measurements and Accuracy was compared. In addition to verification of simulation and experimental theory, research was conducted through actual field cases, and methodologies for crack detection and depth evaluation for concrete structures were presented, and furthermore, the expected effects of improving the soundness and safety of structures were shown.

"The most important factors in concrete crack inspection are crack width, length and 40 depth." the width of the crack may be of low importance, while the direction of the crack defect is more important.

It is agreed that for cracks in structures, the depth and direction of cracks are more important than the width of cracks. However, it is also true that width also affects cracks in concrete cracks. H. Z. Lopez-Calvo et al., M. Ghalehnovi et al., B. Goszczynska et al., D. Z. Yankelevsky et al. Research on the width of cracks has been conducted until recently. References have been added to the text.

“How to determine the crack length in this paper?”

The method to determine the crack length in this study is the method using the UPV method for concrete blocks. The ultrasonic velocity was measured by separating the distance between the sender and the receiver. In a concrete block with 45 mm and 70 mm cracks, the oscillator was positioned 150 mm and 270 mm apart from the crack, and the propagation time (TOF) of the pulse was measured. The propagation time and speed of ultrasonic pulses were checked on the block surface up to a distance of 540 mm, the maximum range that can be measured at intervals of 60 mm. In order to improve the reproducibility and accuracy of the experimental results, the ultrasonic pulse propagation time was measured a total of 5 times for each condition, and the average value was used to apply the known relational expression to evaluate the depth of the crack.

Reviewer 2 Report

The manuscript presents the experimental and numerical results for defect detection and characterization in concrete based on FEM and ultrasonic techniques. The following comments are suggested to be considered by the authors: 

1-Globally, the English level is not sufficient, and some words/sentences are odd or unsound: this article must be proofread by a native speaker.

2- The authors should cite the name of the software used in numerical study?

3- Page 5-Line 178: remove "This is a table".

4-Why the authors select this type of concrete in this study?

5-Why the authors assume that the ultrasonic time of flight (TOF), which increases proportionally as the distance between the transducers increases when there are no cracks, increases as the ultrasonic wave travels through the cracks.?

6- What do authors mean by the word "oscillator"?

7- The first conclusion should be rewritten, it is not understood.

Author Response

Authors feel grateful for the invaluable comments from the reviewer.

1-Globally, the English level is not sufficient, and some words/sentences are odd or unsound: this article must be proofread by a native speaker.

English proofreading was carried out throughout the document.

2- The authors should cite the name of the software used in numerical study?

The following sentence has been added at the beginning of the section3.

“For this, a concrete block with different crack depths was modeled by Onscale is the cloud engineering simulation platform., and the easiness of ultrasonic pulse velocity inspection using two different frequencies was compared.”

Figure 3 title has been corrected as follow:

Figure 3. No.1 (70mm crack) Onscale simulation result image.”

3- Page 5-Line 178: remove "This is a table".

It has been deleted.

4-Why the authors select this type of concrete in this study?

It can be easily confirmed near the beach with the examples of cracks and the fragile environment of concrete mentioned in the introduction. In particular, even in the case of high-risk facilities such as nuclear power plants that are mainly built near the beach, buildings with thick concrete are used without exception. In the case of concrete, it is a kind of porous material, and because of this, it is difficult for ultrasonic waves to penetrate the thick concrete for a long distance. Even today, various factors such as salt penetration, rapid changes in internal and external pressure or temperature, and radioactive materials seriously affect not only concrete but also the internal metal structures involved, threatening structural safety. Therefore, it is necessary to conduct research on concrete used in such high-risk facilities. Relevant references have been added to the introductory section.

5-Why the authors assume that the ultrasonic time of flight (TOF), which increases proportionally as the distance between the transducers increases when there are no cracks, increases as the ultrasonic wave travels through the cracks.?

As shown in Figure 3, if there is no crack, the ultrasonic wave is propagated to the place where receiver 1 is located without bypassing it. However, if there is a crack in the path where the ultrasonic wave propagates, it can be confirmed that the ultrasonic wave is propagated while bypassing the crack. Therefore, the increase in TOF is due to the crack.

6- What do authors mean by the word "oscillator"?

The word “oscillator” has been changed to “transmitter”

7- The first conclusion should be rewritten, it is not understood.

The conclusion part has been modified as follows.

“Finite element analysis and experiments were performed on non-cracked concrete blocks and 45 mm and 70 mm vertical cracks to estimate the crack depth in concrete using ultrasonic propagation time. As a result of measuring the time-of-flight change by changing the positions of the transmitter and receiver converters, it was confirmed that the finite element analysis results agree well with the experimental results. The depth of the defect was calculated using various formulas using the measured values, and the accuracy was compared to show a high degree of similarity to the theoretical value. It also shows the results of applying the verified UPV method to cracks found in the actual field.

In this study, the theoretical crack length measurement method was verified through FEM modeling and actual experiments. In addition, it was confirmed that the crack detection and depth evaluation method using ultrasonic pulse velocity in cracked concrete structures can be a very useful tool to maintain the soundness and safety of the structure.”

Reviewer 3 Report

Article Title: Defect Detection and Characterization in Concrete Based on FEM and Ultrasonic Techniques

This original research article presents non-destructive and modeling-based approaches to characterize and detect cracks in concrete. The findings of this study would promote accuracy in the field of structural health monitoring and such techniques can also be used for other similar structural elements. I think this interesting study deserves publication after few changes:

1)     The abstract contains limited information about the scope of this research.

2)     In abstract include the factors affecting the detection of cracks.

3)     Are these techniques suitable for all types of strength classes of concrete.

4)     Line 62, the reference style not matching the MDPI guidelines.

5)     Line 74, the reference is not according to the standard style.

6)     The writing errors must be rectified.

7)     More details about the design of concrete/structural sample must be added into section 2.

8)     Why authors choose this specific FEM model, what are the limitations.

9)     Figure 3, what different colors in the figures represent must be included in the same figure , a color bar for example.

10)  Figure 4 a and b not helpful at all, must be improved in terms of quality and axis labelling.

11)  Table styles must be consistent with the journal format.

12)  The conclusions must be short in at least 2-3 paragraphs, where each paragraph covers a different aspect.

Author Response

Authors feel grateful for the invaluable comments from the reviewer.

1)     The abstract contains limited information about the scope of this research.

2)     In abstract include the factors affecting the detection of cracks.

1)&2) The abstract has been modified as follows.

- The abstract was revised as follows.

In order to estimate the crack depth in concrete using time-of-flight, finite element analysis and experiments were performed on non-cracked concrete blocks and 45 mm and 70 mm vertical cracks. As a result of measuring the time-of-flight change by changing the positions of the transmitter and receiver, it was confirmed that the finite element analysis results agree well with the experimental results, and various formulas for calculating the depth of defects using the obtained experimental measurements by comparison, high accuracy was confirmed. In addition to verification of simulation and experimental theory, research was conducted through actual field cases, and methodologies for crack detection and depth evaluation for concrete structures were presented, and furthermore, the expected effects of improving the soundness and safety of structures were shown.

3)     Are these techniques suitable for all types of strength classes of concrete.

It is necessary to change the frequency selection or ultrasonic signal analysis depending on the composition of the concrete, but it is expected that all types of concrete structures can be inspected.

4)     Line 62, the reference style not matching the MDPI guidelines.

 It has been changed as follow:

“M. Sansalone et al. proposed a technique for ~ when determining the UPV [13].”

5)     Line 74, the reference is not according to the standard style.

It has been changed as follow:

“N. M. Sutan et al. conducted a comparative experiment of direct and indirect methods of ultrasonic pulse velocity to detect concrete defects. [15]”

6)     The writing errors must be rectified.

11)  Table styles must be consistent with the journal format.

6)&11) English proofreading was carried out throughout the document. And all table styles have been changed.

7)     More details about the design of concrete/structural sample must be added into section 2.

The information of modeling structure and concrete specimen is shown at section 3 and section 4.

8)     Why authors choose this specific FEM model, what are the limitations.

We can use any FEM program, and there is no particular reason to use Onscale. However, the Onscale program is a very useful and specialized finite element analysis program in ultrasonic analysis. The meshing operation is fully automated, allowing the user to focus on the simulation problem at hand. It can also solve static problems that other CAD programs can interpret.

9)     Figure 3, what different colors in the figures represent must be included in the same figure , a color bar for example.

The acoustic pressure has been added at Figure3.

10)  Figure 4 a and b not helpful at all, must be improved in terms of quality and axis labelling.

Figure 4 has been changed.

12)  The conclusions must be short in at least 2-3 paragraphs, where each paragraph covers a different aspect.

The conclusion has been changed as follow:

“Finite element analysis and experiments were performed on non-cracked concrete blocks and 45 mm and 70 mm vertical cracks to estimate the crack depth in concrete using ultrasonic propagation time. As a result of measuring the time-of-flight change by changing the positions of the transmitter and receiver converters, it was confirmed that the finite element analysis results agree well with the experimental results. The depth of the defect was calculated using various formulas using the measured values, and the accuracy was compared to show a high degree of similarity to the theoretical value. It also shows the results of applying the verified UPV method to cracks found in the actual field.

In this study, the theoretical crack length measurement method was verified through FEM modeling and actual experiments. In addition, it was confirmed that the crack detection and depth evaluation method using ultrasonic pulse velocity in cracked concrete structures can be a very useful tool to maintain the soundness and safety of the structure.”

Reviewer 4 Report

Kim et al., have presented the manuscript with title: Defect Detection and Characterization in Concrete Based on FEM and Ultrasonic Techniques, where authors have described the investigation on the method of detecting vertical cracks and determining the depth of cracks through the ultrasonic propagation time increasing through the cracks. Authors have presented the work with good details and presentation of results is also good. I would like to see this article publish but few suggestions are there to be addressed before as follow;

1.      Abstract is so much concise with few lines of the importance, I suggest the authors to highlight their achievements results (values) to appeal the readers.

2.       In the introduction portion, most of the sentences require references,

Line 21-30

Line 34-45 etc. is there some specific reason?

3.      In the introduction, authors should mention the reason (problem statement), why authors have carried out this research and what was lacking in the previous study that they have addressed specifically?

4.      Many sentences are too long and hard to follow. I suggest the authors read through the paper and break long sentences into shorter ones.

5.      There exist several typo and grammatical mistakes in the manuscript, I advise the authors to check the complete manuscript.

6.      For conclusion section I suggest the authors to please concisely summarize the main point of this paper in the conclusion section.

Author Response

Authors feel grateful for the invaluable comments from the reviewer.

  1. Abstract is so much concise with few lines of the importance, I suggest the authors to highlight their achievements results (values) to appeal the readers.

 - The abstract was revised as follows.

In order to estimate the crack depth in concrete using time-of-flight, finite element analysis and experiments were performed on non-cracked concrete blocks and 45 mm and 70 mm vertical cracks. As a result of measuring the time-of-flight change by changing the positions of the transmitter and receiver, it was confirmed that the finite element analysis results agree well with the experimental results, and various formulas for calculating the depth of defects using the obtained experimental measurements by comparison, high accuracy was confirmed. In addition to verification of simulation and experimental theory, research was conducted through actual field cases, and methodologies for crack detection and depth evaluation for concrete structures were presented, and furthermore, the expected effects of improving the soundness and safety of structures were shown.

  1. In the introduction portion, most of the sentences require references,

Line 21-30

Line 34-45 etc. is there some specific reason?

It can be easily confirmed near the beach with the examples of cracks and the fragile environment of concrete mentioned in the introduction. In particular, even in the case of high-risk facilities such as nuclear power plants that are mainly built near the beach, buildings with thick concrete are used without exception. In the case of concrete, it is a kind of porous material, and because of this, it is difficult for ultrasonic waves to penetrate the thick concrete for a long distance. Even today, various factors such as salt penetration, rapid changes in internal and external pressure or temperature, and radioactive materials seriously affect not only concrete but also the internal metal structures involved, threatening structural safety. Relevant references have been added to the text.

  1. In the introduction, authors should mention the reason (problem statement), why authors have carried out this research and what was lacking in the previous study that they have addressed specifically?

The text at the end of the introduction has been amended as follows.

“However, these methods have the disadvantage of having a large error even when tested in theory or measured in an industrial field. This study aimed at the following items. (1) Simulation using the UPV method for concrete blocks as a new approach that could not be confirmed in previous studies. (2) To compare and verify the accuracy of measurements obtained through theoretical and practical experiments. (3) Conduct research through actual field cases as well as verification of simulation and experimental theory (4) Present methodologies for crack detection and depth evaluation for concrete structures and confirm their applicability in actual sites.”

  1. Many sentences are too long and hard to follow. I suggest the authors read through the paper and break long sentences into shorter ones.

Long sentences have been shortened.

  1. There exist several typo and grammatical mistakes in the manuscript, I advise the authors to check the complete manuscript.

A comprehensive review of the documentation has been made.

  1. For conclusion section I suggest the authors to please concisely summarize the main point of this paper in the conclusion section.

The conclusion part has been modified as follows.

“Finite element analysis and experiments were performed on non-cracked concrete blocks and 45 mm and 70 mm vertical cracks to estimate the crack depth in concrete using ultrasonic propagation time. As a result of measuring the time-of-flight change by changing the positions of the transmitter and receiver converters, it was confirmed that the finite element analysis results agree well with the experimental results. The depth of the defect was calculated using various formulas using the measured values, and the accuracy was compared to show a high degree of similarity to the theoretical value. It also shows the results of applying the verified UPV method to cracks found in the actual field.

In this study, the theoretical crack length measurement method was verified through FEM modeling and actual experiments. In addition, it was confirmed that the crack detection and depth evaluation method using ultrasonic pulse velocity in cracked concrete structures can be a very useful tool to maintain the soundness and safety of the structure.”

Round 2

Reviewer 1 Report

The paper is well revised and can be published.